# Morphology of the Antennal Sensilla of the Nymphal Instars and Adults in *Notobitus meleagris* (Hemiptera: Heteroptera: Coreidae)

**DOI:** 10.3390/insects14040351

**Published:** 2023-04-01

**Authors:** Wenli Zhu, Lin Yang, Jiankun Long, Zhimin Chang, Yinlin Mu, Zhicheng Zhou, Xiangsheng Chen

**Affiliations:** 1Institute of Entomology, Guizhou University, Guiyang 550025, China; 2The Provincial Special Key Laboratory for Development and Utilization of Insect Resources, Guizhou University, Guiyang 550025, China; 3The Provincial Key Laboratory for Agricultural Pest Management of Mountainous Regions, Guizhou University, Guiyang 550025, China

**Keywords:** antennae, bug, SEM, ultrastructure

## Abstract

**Simple Summary:**

In this paper, we used scanning electron microscopy to describe the morphological types, number of sensilla, and their distributions on the antennae of each nymphal instar and adult of *Notobitus meleagris* (Hemiptera: Heteroptera: Coreidae). The results show that there were eight subtypes of sensilla on the antennae of the nymphs and 11 subtypes of sensilla on the antennae of the adults. With the growth of instars, the type, quantity, and size of the sensilla gradually increased. Sexual dimorphism did not appear in the type of sensilla but existed in the length and diameter of some sensilla. In addition, we also discussed the functions of different types of sensilla through previous published studies, which will be helpful for further research on the behavioral and biological control of *N. meleagris*.

**Abstract:**

The bamboo bug *Notobitus meleagris* (Fabricius, 1787) is a serious pest of bamboo shoots in China, India, Myanmar, Vietnam, and Singapore. The antennae of the nymphal instars and adults of *N.meleagris* are involved in communication among individuals and finding the host plants. In order to understand the morphology of antennal sensilla, their types, and the distribution of sensilla on the antennae of nymphal instars and adults in *N. meleagris*, we studied the morphology of antennal sensilla with a scanning electron microscope. The antennae of the nymphs and adults comprised the scape, pedicel, and two flagellomeres. Four types and eight subtypes of sensilla were identified in the nymphal instars (sensilla trichodea [St].1, St.2, St.3, sensilla basiconica [Sb].1, Sb.2, sensilla chaetica [Sc].1, Sc.2, sensilla coeloconica [Sco].1), whereas those of the adults had five types and eleven subtypes of sensilla (St.1, St.2, St.3, Sb.1, Sb.2, Sb.3, Sc.1, Sc.2, Sco.1, Sco.2, and sensilla campaniformia [Sca]). There are significant differences in the number, type, and size of the sensilla in different nymphal instars, which increases with the increase in nymphal instars. There was no sexual dimorphism observed in the adult sensilla; however, the length and diameter of St.3, Sb.2, and Sb.3 were sexually dimorphic. The potential functions of each sensillum were discussed based on the morphology and distribution of the antennal sensilla and were compared with similar published studies. Our results provide primary data for further research on the behavioral mechanism, green prevention, and control of *N. meleagris.*

## 1. Introduction

*Notobitus meleagris* (Fabricius) (Hemiptera: Coreidae) is an important pest of bamboo shoots and is distributed in China (Guizhou, Yunnan, Sichuan, Zhejiang, Fujian, Guangdong, Guangxi, Taiwan, and Jiangxi), India, Myanmar, Vietnam, and Singapore [1,2]. There are two–three generations of *N. meleagris* in a year, and they overwinter in weeds or bark as adults. The occurrence period is in line with the bamboo shoot growth period. In April, the adults of the overwintering generation begin to move [3,4]. Bamboo is a perennial evergreen grass that is essential for terrestrial forest ecosystems. It is widely used in the construction, ornamental, and food industries, with high economic, ecological, and social benefits. Bamboo extracts have excellent properties, such as antifree-radical, antioxidative, antiaging, antibacterial, and insecticidal activities. Moreover, they help in regulating blood lipid levels, protecting cardiovascular and cerebrovascular biological activities, and exerting pharmacological effects [5]. There are >120 genera and >1500 species of bamboo worldwide, with 762 species under 44 genera known to occur in China [6]. With the development of the bamboo industry and an increase in planting areas, the extent and degree of damage caused by *N. meleagris* are gradually increasing. In severe cases, >20 individuals are simultaneously present on each bamboo shoot (Figure 1). The overlapping generation of *N. meleagris* significantly affects the health of plants by continuous feeding of both nymphs and adults, and these often accumulate through several generations; this is because the nymphs and adults cause harm by piercing and sucking bamboo shoots, and the feeding site often presents a water stain pattern and has a pungent smell, resulting in the drying or decay of bamboo and seriously affecting its production in the following year [4]. Various studies have been conducted on the habitats of *N. meleagris*; however, the morphology and different nymphal instars have not been clearly described [7,8,9]. Other studies have revealed that *N. meleagris* can be primarily controlled by regulating the use of chemical insecticides [10,11]. This can lead to several problems, such as pesticide resistance, pesticide residue accumulation, and pest resurgence.

Insect sensilla are specialized epidermis structures that mediate an insect’s perception of various environmental stimuli. Insect antennae have numerous sensilla that play an essential role in host localization, habitat selection, companion recognition, foraging, mating, and other processes [12,13]. For example, *Paussus favieri* relies on antennal sensilla and glandular activity to complete communication with host ants [14]. Hygroreceptive sensilla can sense ambient temperature and affect the reproductive behavior and geographic distribution of insects [15]. Recently, the application of biological control and sex pheromones has become increasingly important [16,17,18,19,20]. Many scholars have studied the antennal sensilla of Heteroptera [21,22,23,24,25,26,27,28,29,30,31,32,33,34,35,36,37,38,39,40,41].

During the investigation, we found that *N. meleagris* is highly sensitive to the external environment, prefers to live in shady regions of bamboo shoots, and is highly evasive, thus transmitting information through high-frequency vibrating antennae and moving immediately in response to danger. The morphology of the antennae, including the size and distribution of the antennae sensilla in each nymphal instar and adult, are significantly different. In this study, we redescribed the complex morphological characters of both the adults and nymphal instars of *N. meleagris*. We systematically examined the type, size, and distribution of the antennal sensilla in the adults and nymphal stages of *N. meleagris* via scanning electron microscopy (SEM). Our findings provide basic information regarding the sensory-communicative system of *N. meleagris* nymphs and adults, their behavioral mechanisms associated with biological control, and the host selection and management of *N. meleagris*.

## 2. Materials and Methods

### 2.1. Insects Collection

Ten individual adults with each nymphal instar of *N. meleagris* (Figure 2) were collected from *Bambusa emeiensis* along a river site at Baiziqiao (107.52°E, 26.25°N), Duyun City, Guizhou Province, on 29 August 2021.

### 2.2. Observation and Photography

The photographs of the morphological characteristics of the antennae in the adults and nymphs were acquired using a Canon camera (5D Mark IV, Oitaken, Japan) equipped with a Canon EF 100 mm F/2.8L IS USM Macro Lens, with Godox MF12 Twin Flash used as a light source. Antennae were individually photographed under an ultra-depth-of-field 3D microscope (Keyence 1000, Osaka, Japan), and the pest parameters of each instar were determined.

### 2.3. Sample Preparation for SEM

We anesthetized *N. meleagris* adults and nymphs (five male and female adults each and five individuals of each nymphal instar) at 4 °C for 24 h. The antennae were carefully detached from the head using tiny-tipped forceps and were fixed in 2.5% glutaraldehyde for 24 h. After fixing, ultrasonic cleaning was performed for 15 min. The samples were rinsed thrice with phosphate buffer (0.1 mol/L, pH = 7.4; 15 min of each rinse). Serial dehydration was performed with 30%, 50%, 70%, 80%, 90%, and 100% ethanol; the samples were dehydrated at each concentration for 20 min and then at 100% for 1 day. After dehydration, the samples were dried in an electric blast dryer at 40 °C for 6 h and placed on the sample holder with conductive adhesive based on different observation surfaces. Finally, ion sputtering gold plating was performed for 2 min. The shape and number of sensilla on the entire antennae and segments of the male and female adults, as well as each nymphal instar, were photographed using SEM (JCM6000, JEOL, Tokyo, Japan), and the length of each sensillum was measured.

### 2.4. Statistical Analyses

Scaler software (JCM6000, JEOL, Tokyo, Japan), associated with the SEM system, was used to measure the length of each sensillum and identify and enumerate the antennae of the adults and nymphs. The identification and naming of the antennal sensilla were performed as previously described by Schneider and Zacharuk [42,43]. The description of the shape, socket, surface, and tip of the antennal sensilla referred to the previous research on Heteroptera antennae [24,26,27]. The data and significance of sexual dimorphism were analyzed using student *t*-tests in SPSS 22.0 software (IBM, New York, USA) and expressed as mean ± standard error. Kolmogorov–Smirnov tests were used to assess the normality of the data, and Levene tests were used to determine the homogeneity of variance. Data conforming to the normal distribution and showing homogeneity of variance were analyzed using one-way analysis of variance, and Tukey’s method was used for multiple comparisons.

## 3. Results

### 3.1. Morphological Characteristics of Antennae in Nymphs and Adults

The antennae of *N. meleagris* comprised the scape, pedicel, and two flagellomeres. The total antennal length of the nymph and adult was measured between 3422.86–16587.89 μm (Appendix A). The first instar had clavate antennae, and the fourth segment was more dilated and longer than the first three segments (Figure 3). The length and diameter of the four segments of the first instar were 541.27–1549.81 and 82.29–112.29 μm, respectively (Appendix A). The antennae of the other nymphs and adults showed filiform morphology (Figure 3). Further, the length and diameter of four segments of the second, third, fourth, and fifth instar were 929.02–2164.64 and 80.41–135.17, 1429.59–2870.83 and 122.89–184.16, 2199.84–3828.13 and 174.78–257.00, and 2886.38–4736.79 and 241.58–352.39 μm, respectively. The length and diameter of the four segments of the male and female adults were 3073.63–5783.51 and 236.84–389.80 and 3360.61–5331.14 and 238.47–371.29 μm, respectively (Appendix A). There are many microtrichia densely distributed in the internode membrane of each antennal segment in the adults (Figure 4).

### 3.2. Types and Distribution of Sensilla in the Nymphal Instars

The antennae of each nymphal instar were observed via SEM, which revealed four types of sensilla: sensilla trichodea (St), sensilla basiconica (Sb), sensilla chaetica (Sc), and sensilla coeloconica (Sco). Based on the differences in size and external morphological characteristics, St was divided into three subtypes (St.1, St.2, and St.3), Sb was divided into two subtypes (Sb.1 and Sb.2), and Sc was divided into two subtypes (Sc.1 and Sc.2). Each sensillum showed various distributions among the nymphal instars.

#### 3.2.1. Sensilla Trichodea

Among the three St subtypes, St.1 was the most abundant on the antennal surface in each nymphal instar and was distributed in the second flagellomere. The number of St.1 on the antennae of the first to fifth nymphal instars was 231.0–1226.3 (Table 1). They are long, thin hairs that are curved with a smooth surface and a rounded tip with a flexible socket (Figure 5). The length and base diameter of St.1 in the first to fifth nymphal instars were 30.443–59.548 and 1.362–3.184 μm, respectively. The length of St.1 in the fifth and fourth nymphal instars was significantly greater than that in the third and second nymphal instars. The length of St.1 in the third and second nymphal instars was significantly greater than that in the first nymphal instar. The diameter of St.1 in the fifth nymphal instar did not differ significantly from that in the fourth nymphal instar. The diameter of St.1 in the fourth nymphal instar did not differ significantly from that in the third and below nymphal instars. However, the diameter of St.1 in the fifth nymphal instar was significantly greater than that in the third and below nymphal instars (Appendix A). Among the three St subtypes, St.2 had the shortest length and was distributed vertically in the scape, pedicel, and first flagellomere of the fourth and fifth nymphal instars. The number of St.2 on the fourth and fifth nymphal instar antennae was 2.0–4.9 (Table 1). They are short and slightly curved, with smooth surfaces and a rounded tip, and a flexible socket. (Figure 5). The length and basal diameter of St.2 in the fourth and fifth nymphal instars were 27.148–41.575 and 2.770–4.526 μm, respectively. The length of St.2 in the fifth nymphal instar was significantly greater than that in the fourth nymphal instar, whereas no significant difference was observed in the diameter of St.2 in these nymphal instars (Appendix A). St.3 was distributed vertically in the scape, pedicel, and first flagellomere of the fourth and fifth nymphal instars. The number of St.3 on the antennae of the fourth and fifth nymphal instars was 3.1–8.0 (Table 1). They are long, thin hairs that are nearly straight and have a smooth surface, with shrinkage and a softly rounded tip and a flexible socket. (Figure 5). The length and base diameter of St.3 in the fourth and fifth nymphal instars were 59.228–69.244 and 4.004–5.117 μm, respectively. The length and diameter of St.3 did not differ significantly between the fifth and fourth nymphal instars (Appendix A).

#### 3.2.2. Sensilla Basiconica

Among the two Sb subtypes, Sb.1 was distributed vertically in the second flagellomere of all the nymphal instars. The number of Sb.1 on the first to fifth nymphal instar antennae was 8.6–43.1 (Table 1). They are short, with a longitudinally grooved wall, a blunt and apertured tip, and an inflexible socket (Figure 6). The length and base diameter of Sb.1 in the first to fifth nymphal instars were 6.929–12.822 and 1.215–3.570 μm, respectively, and differed significantly (Appendix A). Sb.2 was vertically distributed in the second flagellomere of the third to fifth nymphal instars. The number of Sb.2 on the antennae of the third to fifth nymphal instars was 4.4–17.6 (Table 1). They are short and thick with a smooth surface, a blunt tip, and an inflexible socket. (Figure 6). The length and base diameter of Sb.2 in the third to fifth nymphal instars were 34.193–43.469 and 3.992–5.659 μm, respectively. No significant differences were noted in the length of Sb.2 in all the nymphal instars, but the diameter of Sb.2 in the fifth nymphal instar was significantly greater than that in the third nymphal instar (Appendix A).

#### 3.2.3. Sensilla Chaetica

Among the two Sc subtypes, Sc.1 was strictly distributed in the scape, pedicel, and first flagellomere of all the nymphal instars. The number of Sc.1 on the antennae of the first to fifth nymphal instars was 90.9–552.8 (Table 1). They are long and straight with a longitudinally grooved wall, a sharp tip, and a flexible socket. (Figure 7). The length and base diameter of Sc.1 in the first to fifth nymphal instars were 52.070–155.440 and 3.943–8.474 μm, respectively. The length of Sc.1 in the fifth and fourth nymphal instars was significantly greater than that in the third and second nymphal instars. The length of Sc.1 in the first nymphal instar was significantly smaller than that in the third nymphal instar. The diameter of Sc.1 in the fifth nymphal instar was significantly greater than that in the fourth and below nymphal instars. The diameter of Sc.1 in the fourth and first nymphal instars was significantly smaller than that in the other nymphal instars (Appendix A). Sc.2 was strictly distributed in the second flagellomere of all the nymphal instars. The number of Sc.2 on the antennae of the first to fifth nymphal instars was 5.0–22.9 (Table 1). They are long and straight with a longitudinally grooved wall, a rounded tip, and a flexible socket. (Figure 7). The length and base diameter of Sc.2 in the first to fifth nymphal instars were 34.763–79.512 and 2.098–7.392 μm, respectively. The length of Sc.2 in the fifth nymphal instar was significantly greater than that in the fourth nymphal instar. The length of Sc.2 in the fourth nymphal instar was significantly greater than that in the first to third nymphal instars. The diameter of Sc.2 decreased significantly in descending order from the fifth to the first instar nymphal stages (Appendix A).

#### 3.2.4. Sensilla Coeloconica

Sco.1 was distributed vertically in the second flagellomere of all the nymphal instars. The number of Sco.1 on the antennae of the first to fifth nymphal instars was 1.2–19.8 (Table 1). They are short, coniform pegs with a longitudinally grooved wall and a rounded and apertured tip that is located in cavity-shaped sockets. (Figure 8). The length and base diameter of Sco.1 in the first to fifth nymphal instars were 3.193–5.782 and 1.447–2.385 μm, respectively. The length of Sco.1 in the fifth and fourth nymphal instars was significantly greater than that in the second and first nymphal instars, and the diameter of Sco.1 in the fifth nymphal instar was significantly greater than that in the first to fourth nymphal instars (Appendix A).

### 3.3. Types and Distribution of Sensilla in Adults

The antennae of the adults were observed via SEM, which revealed seven types of sensilla: St, Sb, Sc, Sco, and sensilla campaniformia (Sca). Based on the differences in size and external morphological characteristics, St was divided into three subtypes (St.1, St.2, and St.3), Sb was divided into three subtypes (Sb.1, Sb.2, and Sb.3), Sc was divided into two subtypes (Sc.1 and Sc.2), and Sco was divided into two subtypes (Sco.1 and Sco.2). These 13 subtypes of sensilla were distributed in the antennae of both the male and female adults; however, the size of these sensilla varied between the sexes.

#### 3.3.1. Sensilla Trichodea

Among the three St subtypes, St.1 was distributed in the second flagellomere of the adults. The number of St.1 on the antennae of the males and females was 1683.7–2058.8 (Table 1). The shape of the adult St.1 was consistent with that of the nymph St.1 (Figure 5). The length and base diameter of St.1 in the adults were 66.431–70.089 and 3.259–3.522 μm, respectively, with no significant difference between the males and females (Appendix A). St.2 was distributed in the second flagellomere of the adults. The number of St.2 on the antennae of the males and females was 7.7–11.0 (Table 1). The shape of the adult St.2 was consistent with that of the nymph St.2 (Figure 5). The length and base diameter of St.2 in the adults were 32.851–36.387 and 2.061–3.348 μm, respectively, with no significant difference between the males and females (Appendix A). St.3 was distributed in the scape, pedicel, and first flagellomere of the adults. The number of St.3 on the antennae of the males and females was 9.9–19.2 (Table 1). The shape of the adult St.3 was consistent with that of the nymph St.3 (Figure 5). The length and base diameter of St.3 in the adults were 53.230–62.081 and 3.408–3.745 μm, respectively. The length of St.3 was significantly greater in the males than in the females; however, there was no significant difference in the diameter of St.3 between the sexes (Appendix A).

#### 3.3.2. Sensilla Basiconica

Among the three Sb subtypes, Sb.1 was distributed in the second flagellomere of the adults. The number of Sb.1 on the antennae of the males and females was 101.5–120.1 (Table 1). The shape of the adult Sb.1 was consistent with that of the nymph Sb.1 (Figure 6). The length and base diameter of Sb.1 in the adults were 12.575–13.948 and 3.061–3.191 μm, respectively. The length of Sb.1 was significantly greater in the males than in the females; however, there was no significant differences in the diameter of Sb.1 between the sexes (Appendix A). Sb.2 was distributed in the second flagellomere of the adults. The number of Sb.2 on the antennae of the males and females was 27.2–50.4 (Table 1). The shape of the adult Sb.2 was consistent with the nymph Sb.2 (Figure 6). The length and base diameter of Sb.2 in the adults were 32.716–36.650 and 6.363–7.503 μm, respectively, with significantly greater values in the males than in the females (Appendix A). Sb.3 was distributed in the scape, pedicel, and first flagellomere of the adults. The number of Sb.3 on the antennae of the males and females was 10.5–14.7 (Table 1). They are short and small, with a smooth wall, a blunt tip, and a flexible socket. (Figure 6). The length and base diameter of Sb.3 in the adults were 14.557–16.730 and 2.788–3.251 μm, respectively. There was no significant difference in the length of Sb.3 between the males and females; however, the diameter of Sb.3 was greater in the females than in the males (Appendix A).

#### 3.3.3. Sensilla Chaetica

Among the two Sc subtypes, Sc.1 was distributed in the scape, pedicel, and first flagellomere of the adults. The number of Sc.1 on the antennae of the males and females was 737.6–960.1 (Table 1). The shape of the adult Sc.1 was consistent with that of the nymph Sc.1 (Figure 7). The length and base diameter of Sc.1 in the adults were 120.299–129.575 and 9.463–10.328 μm, respectively, with no significant differences between the males and females (Appendix A). Sc.2 was distributed in the second flagellomere of the adults. The number of Sc.2 on the antennae of the males and females was 112.6–166.1 (Table 1). The shape of the adult Sc.2 was consistent with that of nymph Sc.2 (Figure 7). The length and base diameter of Sc.2 in the adults were 78.625–87.180 and 7.229–7.855 μm, respectively. The length of Sc.2 was significantly greater in females than in males; however, the diameter of Sc.2 did not differ significantly between the sexes (Appendix A).

#### 3.3.4. Sensilla Coeloconica

Among the two Sco subtypes, Sco.1 was distributed in the second flagellomere of the adults. The number of Sco.1 on the antennae of the males and females was 30.3–40.5 (Table 1). The shape of the adult Sco.1 was consistent with that of the nymph Sco.1 (Figure 8). The length and base diameter of Sco.1 in the adults were 5.202–6.502 and 2.044–2.455 μm, respectively, with significantly higher values in the males than in the females (Appendix A). Sco.2 was sporadically embedded in the scape of the adults. The number of Sco.2 on the antennae of the males and females was 5.1–8.9 (Table 1). They are short coniform pegs with a smooth wall, located in cavity-shaped sockets. (Figure 8). The length and base diameter of Sco.2 in the adults were 1.791–2.236 and 1.455–1.812 μm, respectively, with no significant difference between the sexes (Appendix A).

#### 3.3.5. Sensilla Campaniformia

Sca was oval-shaped and clustered at the base of the scape in adults. They consist of an oval plate with an ecdysial pore, situated inside a tight columnar socket. (Figure 8). The length and base diameter of Sca in the adults were 9.699–13.247 and 6.292–12.498 μm, respectively. The length of Sca did not differ significantly between the males and females; however, the diameter of Sca was significantly greater in females than in males (Appendix A).

### 3.4. Comparison of Antennal Sensilla in Adults and Nymphal Instars

There were four types and eight subtypes of sensilla on the antennae of the nymphs; St.2 and St.3 were unique to the fourth and fifth nymphal instars, whereas Sb.2 was unique to the third, fourth, and fifth nymphal instars. Except for the abovementioned four sensilla, the other sensilla were shared among all the nymphal instars. Additionally, five types and 11 subtypes of sensilla were identified on the antennae of the adults. There was no sexual dimorphism among these types, and Sb.3, Sco.2, and Sca were unique to the adults.

## 4. Discussion

We identified St.1, Sb.1, Sc.1, Sc.2, and Sco.1 on the antennae of the first and second nymphal instars. The antennae of the third nymphal instar had Sb.2 in addition to the sensilla present on the antennae of the first and second nymphal instars. The antennae of the fourth and fifth nymphal instars had St.2 and St.3 in addition to the sensilla present on the antennae of the third nymphal instar. The antennae of the adults had Sb.3, Sco.2, and Sca, in addition to the sensilla present on the antennae of the fourth and fifth nymphal instars. The subtypes of sensilla in the *N. meleagris* nymphs to adults gradually increased from 5 to 11. The structure and type of these sensilla were similar to those of other Hemiptera insects [28,29], with varying fine structures. The antennal sensilla in *N. meleagris*, particularly in the nymphs, can be divided according to antennal segments. The types of sensilla in the first, second, and third antennal segments were identical, whereas the types of sensilla in the fourth antennal segment differed remarkably from those in the other three segments (Figure 9). The fourth antennal segment of *N. meleagris* had the highest number and density of sensilla, indicating that it plays an essential role in sensing in *N. meleagris*. The length of the antennae increased with the increase in the instar stage, with altered morphology and a gradual increase in the types and number of sensilla. However, *N. meleagris* nymphs lacked Sb.3, Sco.2, and Sca, suggesting that these three sensilla are related to the sexual maturity of *N. meleagris*. There was no sexual dimorphism in the types and numbers of sensilla in *N. meleagris*, except for the length and thickness of St.3, Sb.1, Sb.2, Sb.3, Sc.2, Sco.1, and Sca. It is speculated that these sensilla play an essential role in the mating or oviposition behavior of *N. meleagris*.

The antennae are the principal organs of the insect sensory system. Various sensilla form essential functional elements to screen and select environmental factors and perceive chemical, mechanical, and physical signals [44,45]. Sensilla trichodea of *N. meleagris* is similar in appearance to those described for *Riptortus pedestris* [26], which is common in insects [46,47,48] and is frequently associated with the sense of smell [49], with particular sensitivity to pheromones [50,51]. However, the number and types of sensilla trichodea increase with the increase in instar stage of *N. meleagris*; therefore, the ability to perceive pheromones is stronger in higher instar stages. Additionally, Sb.1 is similar in appearance to those described in *Pyrrhocoris sibiricus* [30], and sensilla basiconica is widely believed to be involved in the perception of volatile plant compounds [52], which often influence the oviposition behaviors of plant-feeding insects [31]. The length and diameter of sensilla basiconica in *N. meleagris* showed sexual dimorphism, which may be related to the oviposition behaviors of the females. The sensilla chaetica of *N. meleagris* is similar in appearance to those described for *Apolygus lucorum* [53], which is the longest and hardest sensilla of *N. meleagris*, and usually protects other receptors [54,55,56]. Furthermore, the sensilla coeloconica of *N. meleagris* is similar in appearance to those described for *Eocanthecona furcellata* [32], which is a sensillum that detects temperature and humidity [57,58]. Sco.1 is present in all *N. meleagris* nymphal instars and is speculated to be the main sensilla through which they sense external temperature and humidity. The sensilla campaniformia of *N. meleagris* is similar in appearance to those described for Gerromorpha [33], found in both the antennae and palate, playing a mechanical role [34,35,59]. Therefore, it is speculated that sensilla campaniformia also plays the role of mechanical reception in *N. meleagris*, that is to say, the mechanical reception of the adult is stronger than that of the nymphs [36]. This also explains why *N. meleagris* adults have a stronger ability to perceive signals than nymphs. The specific functions of different sensillum subtypes should be further verified by conducting detailed electrophysiological studies on single sensilla.

Receptors are the primary means of communication between individual insects and the environment; moreover, insects use sensilla to locate their partners and host plants [37]. In addition, detailed studies on the eggs and nymphs of insects can help understand their taxonomy, biology, and evolution [38,39]. Based on the morphological characteristics of the antennal receptors in each instar, we can determine the sensory structure of the antennae related to pheromones, particularly chemoreceptors, to provide a reference for further studies on the biological and behavioral mechanisms of *N. meleagris* and a basis for blocking the function of sensilla and controlling *N. meleagris* through synthetic pheromones. This could lay a theoretical foundation for the green prevention and control of insects and aid in forestry plant protection.

## Figures and Tables

**Figure 1 insects-14-00351-f001:**
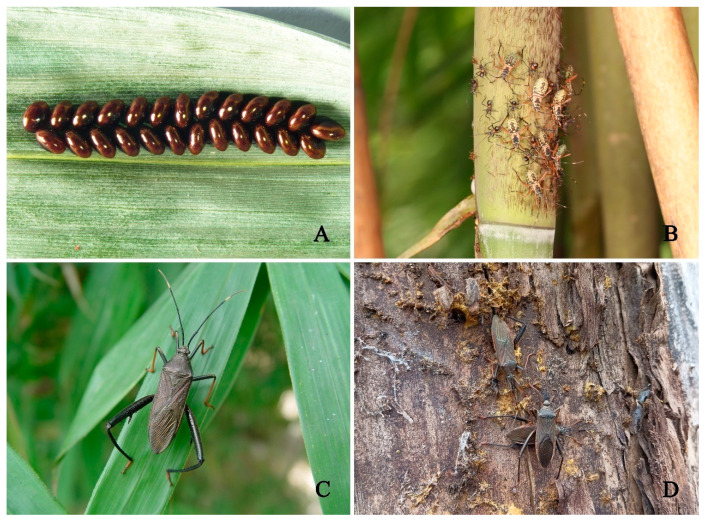
*Notobitus melegaris* (Fabricius). (**A**). Egg mass; (**B**). Nymphal stages; (**C**). Adult; (**D**). Overwinter ecological habitus of adults.

**Figure 2 insects-14-00351-f002:**
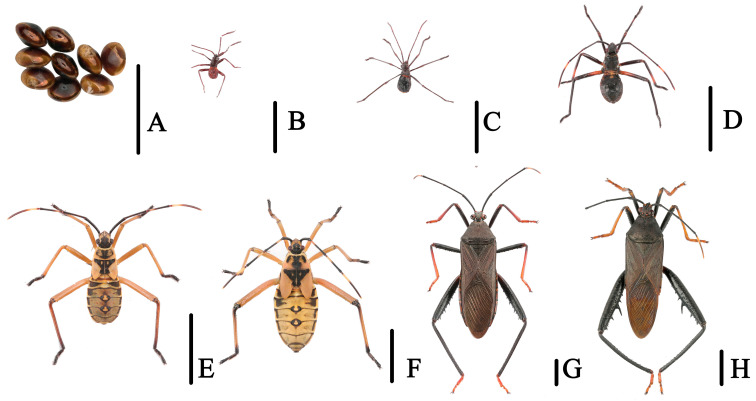
The complete life stages of *Notobitus meleagris* (Fabricius) from eggs to adults. (**A**). Egg mass; (**B**–**F**) First to fifth nymphal instars; (**G**) Adult, female; (**H**) Adult, male. Scale bars = (**A**–**H**) (5000 μm).

**Figure 3 insects-14-00351-f003:**
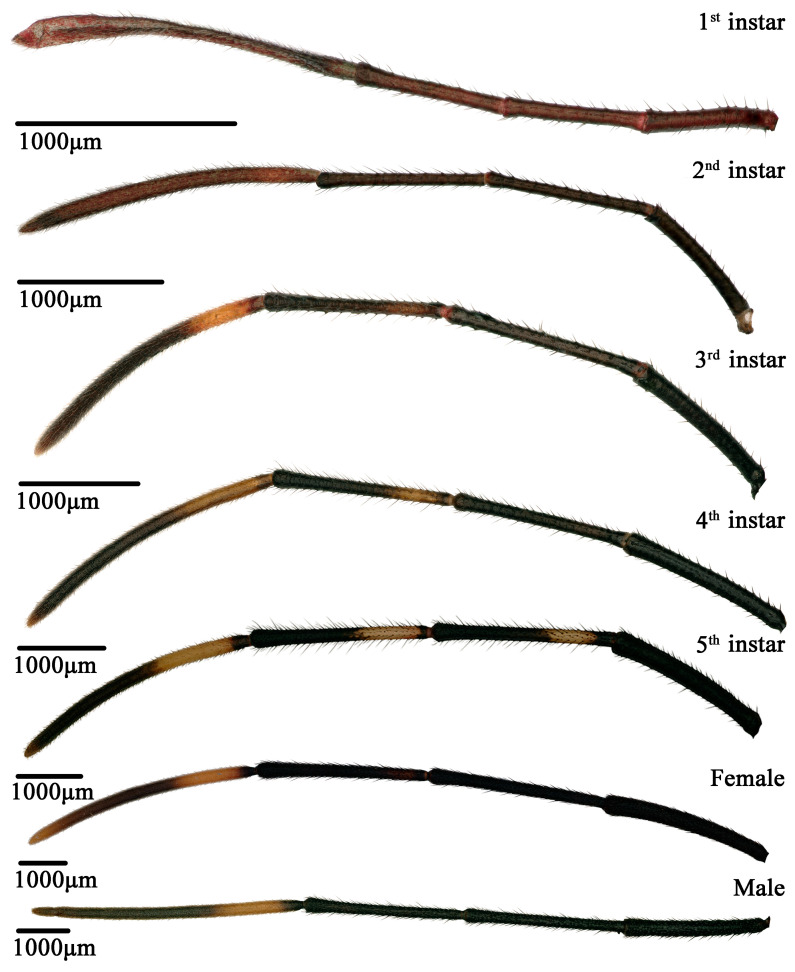
Morphological characteristics of *N. meleagris* antennae at each nymphal instar and adult.

**Figure 4 insects-14-00351-f004:**
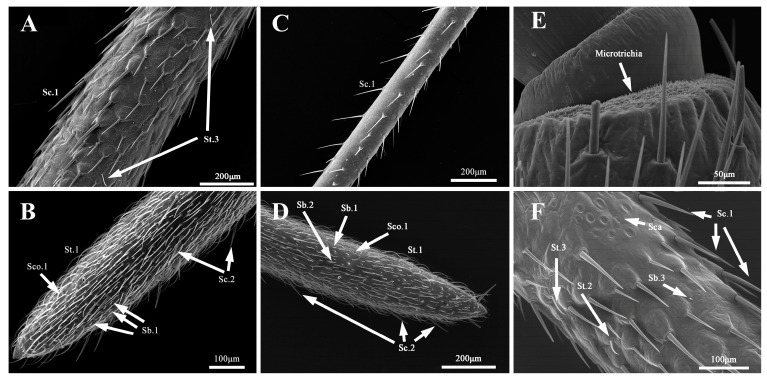
*N. meleagris* (**A**) Scape, pedicel, and first flagellomere of adult; (**B**) second flagellomere of adult; (**C**) scape, pedicel and first flagellomere of nymph; (**D**) second flagellomere of nymph; (**E**) microtrichia; (**F**) base of the scape of antennae of adult.

**Figure 5 insects-14-00351-f005:**
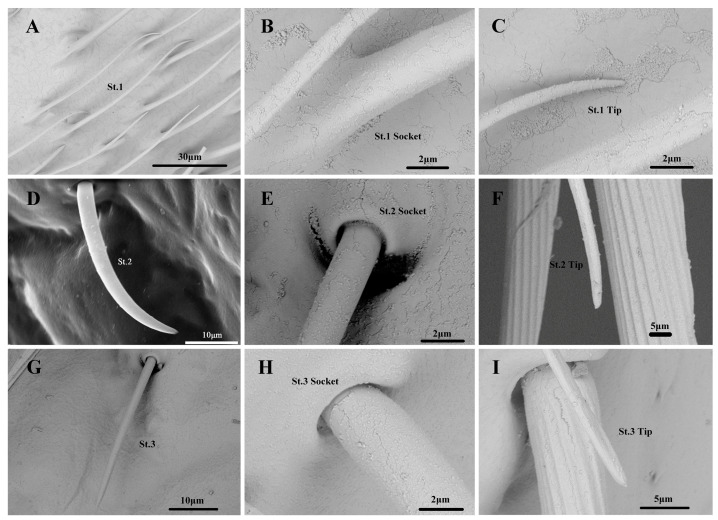
(**A**–**I**). Sensilla Trichodea of *N. meleagris* antennae.

**Figure 6 insects-14-00351-f006:**
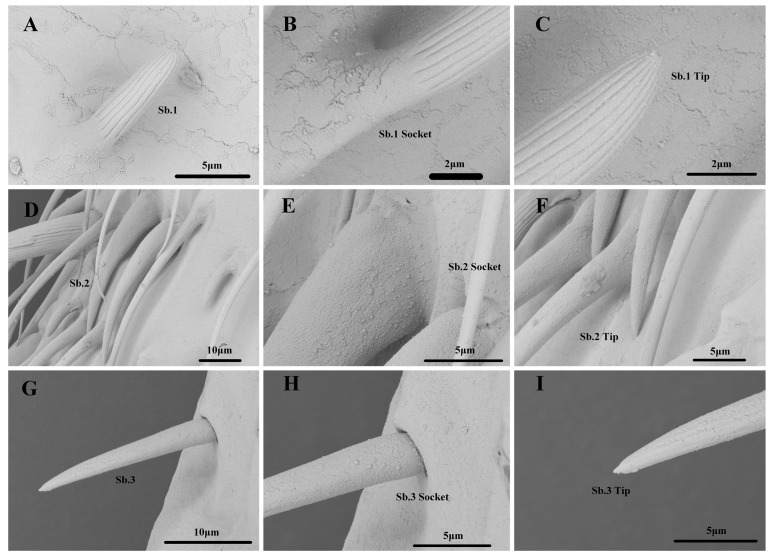
(**A**–**I**). Sensilla Basiconica of *N. meleagris* antennae.

**Figure 7 insects-14-00351-f007:**
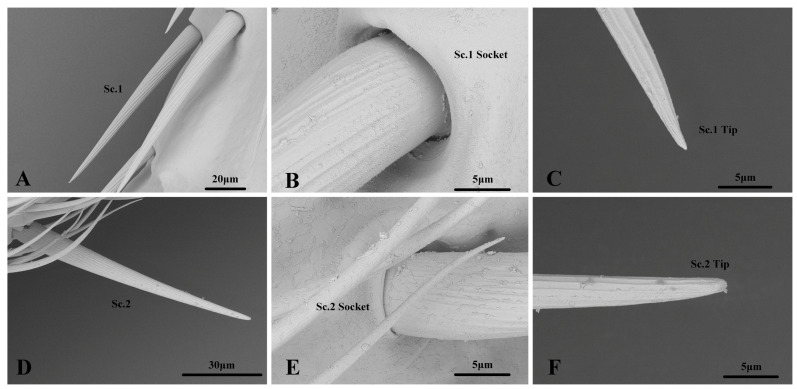
(**A**–**F**). Sensilla Chaetica of *N. meleagris* antennae.

**Figure 8 insects-14-00351-f008:**
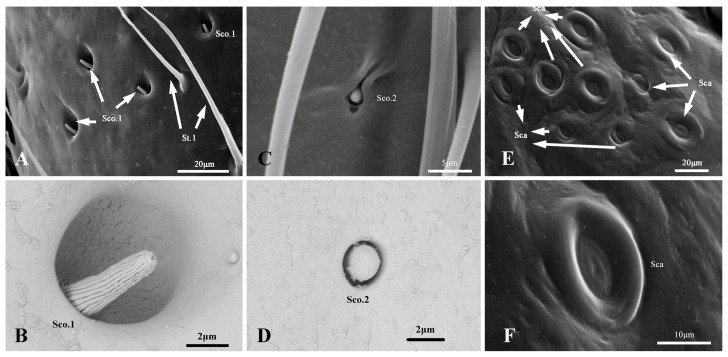
(**A**–**D**). Sensilla Coeloconica of *N. meleagris* antennae; (**E**,**F**). Sensilla Campaniformia of *N. meleagris* antennae.

**Figure 9 insects-14-00351-f009:**
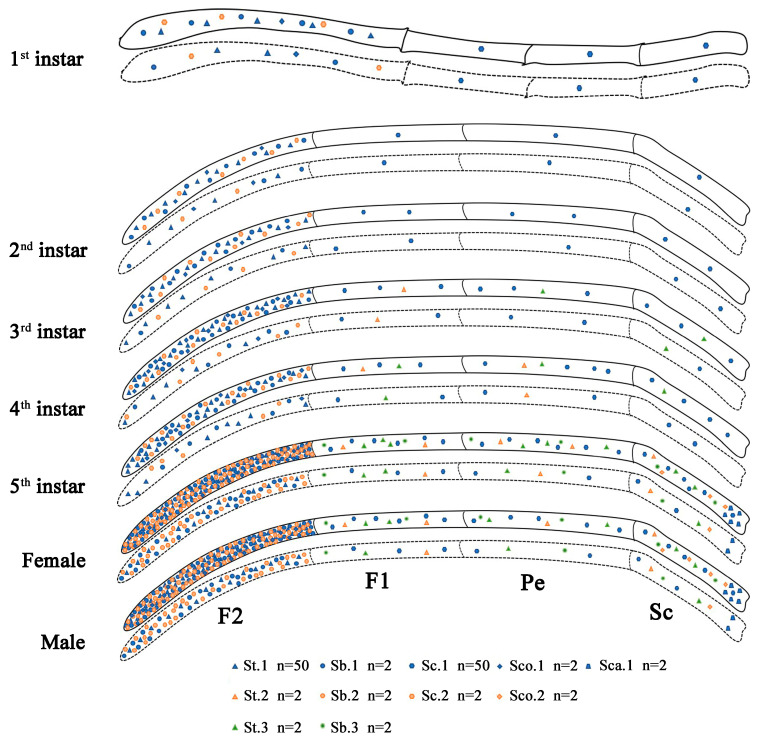
Distribution of antennal sensilla in *N. meleagris* antennae at each instar. The solid and dotted lines represent the dorsal and ventral sides, respectively.

**Table 1 insects-14-00351-t001:** Number and distribution of sensilla in the antennae of *N. meleagris*.

Type/Subtypes of Sensilla (Distribution)	Number of the Sensilla							Degree Freedom	F Value
	1st Star	2nd Star	3rd Star	4th Star	5th Star	Male	Female		
St.1(F2)	249.8 ± 18.773 e	444.6 ± 19.896 de	722.4 ± 24.060 cd	1118.2 ± 667.880 bc	1209.4 ± 16.928 b	1854.6 ± 170.915 a	1889.8 ± 169.043 a	(6, 28)	45.459
St.2(Sc, Pe, F1)	–	–	–	2.0 ± 0.000 bc	4.2 ± 0.735 b	8.6 ± 0.927 a	10.0 ± 1.049 a	(6, 28)	50.277
St.3(Sc, Pe, F1)	–	–	–	4.6 ± 1.470 bc	6.6 ± 1.364 b	12.0 ± 2.145 a	15.8 ± 3.382 a	(6, 28)	14.021
Sb.1(F2)	9.2 ± 0.583 c	17.2 ± 3.216 bc	25.6 ± 2.694 bc	39.2 ± 3.555 b	38.0 ± 5.060 b	110.8 ± 9.303 a	110.4 ± 8.600 a	(6, 28)	58.911
Sb.2(F2)	–	–	5.6 ± 1.208 c	7.8 ± 1.114 c	15.2 ± 2.354 bc	35.6 ± 8.406 ab	40.2 ± 10.180 a	(6, 28)	10.543
Sb.3(Sc, Pe, F1)	–	–	–	–	–	12.6 ± 1.470 a	12.6 ± 2.112 a	(6, 28)	39.970
Sc.1(Sc, Pe, F1)	98.4 ± 7.541 d	155.80 ± 3.323 cd	290.8 ± 41.897 c	439.6 ± 36.214 b	537.0 ± 15.758 b	772.6 ± 34.964 a	906.0 ± 54.109 a	(6, 28)	85.928
Sc.2(F2)	5.8 ± 0.800 b	14.4 ± 1.860 b	14.8 ± 2.083 b	17.2 ± 3.441 b	21.4 ± 1.503 b	127.4 ± 14.807 a	143.2 ± 22.850 a	(6, 28)	32.110
Sco.1(F2)	1.4 ± 0.245 c	6.2 ± 0.663 c	8.8 ± 0.374 bc	18.2 ± 1.562 b	17.0 ± 1.140 b	34.0 ± 2.665 a	35.4 ± 5.134 a	(6, 28)	32.548
Sco.2(Sc)	–	–	–	–	–	8.0 ± 0.894 a	6.4 ± 1.288 a	(6, 28)	35.729
Sca(Sc)	–	–	–	–	–	Some	Some	(6, 28)	2.551

Each row of data is analyzed together, and the letters in the table indicate significant differences (*p* < 0.05, Tukey).

## Data Availability

The data presented in this study are available in the article.

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
