# Peer review of "Morphology of the Antennal Sensilla of the Nymphal Instars and Adults in Notobitus meleagris (Hemiptera: Heteroptera: Coreidae)"

_insects, 2023, doi:10.3390/insects14040351_

Round 1

Reviewer 1 Report (Previous Reviewer 1)

The article provides original data on antennal sensilla on Notobitus meleagris. I suggest the editor accepting the manuscript after a minor revision.

Comments:

As for the references about the research done by other authors on antennal sensilla in Heteroptera - later in the manuscript you cite more work than what you have mentioned in the introduction. I suggest taking all those works and citing them at the beginning when you mention antennal sensilla in Heteroptera, and later re-cite them when you mention something that was in the specific publication. 

Figures 5, 6, 7, 8- which photos are from instars and which from adults? If you compare them, the photos should be described.

I also suggest checking the whole manuscript for spelling/grammar mistakes because I still see a few.

Author Response

Response to Reviewer 1 Comments

Dear anonymous reviewer,

Thank you very much for your comments and professional advice. These opinions help to improve academic rigor of our article. Based on your suggestion and request, we have made corrected modifications on the revised manuscript. We hope that our work can be improved again. Furthermore, we would like to show the details as follows:

Question: As for the references about the research done by other authors on antennal sensilla in Heteroptera - later in the manuscript you cite more work than what you have mentioned in the introduction. I suggest taking all those works and citing them at the beginning when you mention antennal sensilla in Heteroptera, and later re-cite them when you mention something that was in the specific publication.

Response: I've cited all the Hemiptera references of my article in the introduction.

Question: Figures 5, 6, 7, 8- which photos are from instars and which from adults? If you compare them, the photos should be described.

Response: Figure 5, 6, 7, 8 only shows the morphological characteristics of different sensilla subtypes, and the comparative information of nymphal instars and adults is presented in the text, table, and figure, such as line 422, table 1 and figure 9.

Question: I also suggest checking the whole manuscript for spelling/grammar mistakes because I still see a few.

Response: I am very sorry for the spelling and grammar mistakes. We have carefully reviewed the entire manuscript and corrected it.

We would like to thank again for the valuable advice. Look forward to hearing from you soon.

Thank you very much.

With best regards

Zhu Wenli

Reviewer 2 Report (Previous Reviewer 3)

The authors have made many corrections and added suggestions as requested. Therefore, it may be published. 

Author Response

Response to Reviewer 2 Comments

Dear anonymous reviewer,

Thank you very much for your comments and professional advice. These opinions help to improve academic rigor of our article. Based on your suggestion and request, we have made corrected modifications on the revised manuscript. We hope that our work can be improved again. Furthermore, we would like to show the details as follows:

Question: The authors have made many corrections and added suggestions as requested. Therefore, it may be published. 

Response: Thank you very much for your valuable advice, which makes my manuscript more rigorous.

We would like to thank again for the valuable advice. Look forward to hearing from you soon.

Thank you very much.

With best regards

Zhu Wenli

Reviewer 3 Report (New Reviewer)

These are my main comments on the manuscript (insects-2309096) entitled “Morphology of antennal sensilla of the nymphal instars and adults in Notobitus meleagris (Hemiptera: Heteroptera)”. This work investigates the different and distribution of antennal sensilla in this hemipterous pest. Following substantial revisions should be incorporated in the manuscript prior to acceptance.

1. I have concerns about the manuscript sections that I believe need to be addressed in order to improve its clarity.

2. A hypothesis for this study is needed.

3. In results section, statistical data are needed. Please, provide the F value, degree freedom, and p-value obtained from analysis of variance.

4. Other revisions could be checked in PDF attached.

Author Response

Response to Reviewer 3 Comments

Dear anonymous reviewer,

Thank you very much for your comments and professional advice. These opinions help to improve academic rigor of our article. Based on your suggestion and request, we have made corrected modifications on the revised manuscript. We hope that our work can be improved again. Furthermore, we would like to show the details as follows:

Question: I have concerns about the manuscript sections that I believe need to be addressed in order to improve its clarity. 

Response: According to your suggestions, we have adjusted the sentence of the manuscript, hoping that it can make the article clearer.

Question: A hypothesis for this study is needed. 

Response: We proposed a hypothesis based on the sensitivity of the antennal sensilla, their size and distribution in each nymphal instar and adult, and the preference for shady habitat of N. meleagris. Our research provides fundamental understanding of N. meleagris nymphs and adults' sensory-communicative system, their behavioral controls connected to biological control, and the selection of hosts to management for N. meleagris.

Question: In results section, statistical data are needed. Please, provide the F value, degree freedom, and p-value obtained from analysis of variance.

Response: I have added F value, degree freedom, and p-value in table 1, table S1, table S2 and table S3. Thank you for your advice to make my data more complete.

Question: Other revisions could be checked in PDF attached.

Response: We have revised the manuscript according to your suggestions in the PDF.

Question from pdf file

Page 1, line 39

Question: Keywords should be in alphabetic order. Also, keywords serve to widen the opportunity to be retrieved from a database. To put words that already are into title and abstracts makes KW not useful. Please choose terms that are neither in the title nor in abstract.

Response: We have changed the keywords in alphabetical order. Also, we replaced some keywords with synonyms. However, some terminology has no alternative synonym, so they are not modified, such as antennae.

Page 6, line 153

Question: Repetitive information is found in this item. The number and distribution of sensilla appear in table 1. Here I suggest describing the type of sensilla.

Response: In order to better understand the supplementary table, we have provided a brief explanation of the number and distribution of sensilla in this section. I prefer to retain this information to support my supplemental data.

We would like to thank again for the valuable advice. Look forward to hearing from you soon.

Thank you very much.

With best regards

Zhu Wenli

This manuscript is a resubmission of an earlier submission. The following is a list of the peer review reports and author responses from that submission.

Round 1

Reviewer 1 Report

The paper provides valuable data on antennal sensilla of N. meleagris. However, some changes need to be done in order to approve the manuscript:
1. The title should be changed. The article shows morphological, not ultrastructural data.
2. Line 75 - you cannot write that antennal sensilla of Heteroptera were extensively studied and only back it up with two source examples. Much more references are needed and could easily be acquired.
3. Lines 75-82, I am not sure this should be in Introduction. Maybe it is better to move it to Materials and Methods.
4. No description is provided according to which you described the types of sensilla. No morphology information is given. As for their putative function, many details were left out. St, for example, are not responsible for smell, as there is no large amount pores on their surface. They are sometimes associated with the sense of gustation, as contact chemoreceptors with one pore on the tip. And even that is not their main function, because they are mostly mechanoreceptors, therefore perceiving touch. A full list of sensilla types with their morphology and putative function should be given in order to discuss them with your results.

Author Response

Response to Reviewer 1 Comments

Thank you very much for the valuable advice of the anonymous reviewer!

Point 1: The title should be changed. The article shows morphological, not ultrastructural data.

Response 1: I have changed the title.

Point 2: 2. Line 75 - you cannot write that antennal sensilla of Heteroptera were extensively studied and only back it up with two source examples. Much more references are needed and could easily be acquired.

Response 2: I've added references support.

Point 3: Lines 75-82, I am not sure this should be in Introduction. Maybe it is better to move it to Materials and Methods.

Response 3: I've added references support. I have divided it into a new paragraph after Line 75. I think this paragraph is to say the cause and significance of my research, so it should exist in the Introduction, not in the Material and Method.

Point 4: No description is provided according to which you described the types of sensilla. No morphology information is given. As for their putative function, many details were left out. St, for example, are not responsible for smell, as there is no large amount pores on their surface. They are sometimes associated with the sense of gustation, as contact chemoreceptors with one pore on the tip. And even that is not their main function, because they are mostly mechanoreceptors, therefore perceiving touch. A full list of sensilla types with their morphology and putative function should be given in order to discuss them with your results.

Response 4: I have added a list of expected function in Table S2 and Table S3.

Reviewer 2 Report

The manuscript presents high qualities. Its main failing is the confusion of microtrichia with sensilla squamiformia or Böhm sensilla.

Author Response

Response to Reviewer 2 Comments

Thank you very much for the valuable advice of the anonymous reviewer!

Point 1:

Simple summary number to be changed L16,17

Abstract :

L27, two flagellomeres

L29, to delete squamiformia

L31, Böhm sensilla to delete

Introduction: worthwhile prospects

L70, Paussus favieri in italics

L72, Hygroreceptive sensilla

Materials and Methods, nothing to report

Response 1: I have corrected them.

Point 2: Results:

L128, two flagellomeres

L144, to delete sensilla squamiformia

L166, basal diameter

L216, and all in text and figures, write coeloconica instead of coleoconica

L219, remark Fig. 6F: poorly-visible grooves !

L228-234, Sensilla squamiformia : paragraph to delete , and also Fig. 7A,B. Your « sensilla

squamiformia » ar noninnervated microtrichia in the shape of cuticular scales. See Sq in all other insects and reference in Lepidoptera :

Faucheux M.J. 1999. Biodiversity and unity of sensory organs in lepidopteran insects. Société des Sciences Naturelles

de l’Ouest de la France (éd.), Nantes, 296 p.

L311-317, A micrograph of the first antennal segment would be necessary to show the location of sensilla campaniformia because their number is unusual.

L319-322 to delete : Your « Böhm bristles » are noninnervated microtrichia. See Bb in all other

insects and reference (Faucheux 1999).

L322, Bb are no described !

Response 2: I have corrected them. And the Fig. 7D show the first antennal segment to show the location of sensilla campaniformia.

Point 3: Discussion

L336, delete Bb

L338, seenuimbers ?

L348, delte BB

L355, write Sensilla trichodea

L359, write Sensilla basiconica

L362, write Sensilla chaetica

L364, you write : Sco (Sensilla coeloconica) is a porous sensillum that detects temperature and

humidity. It is erroneous because thermo-hygroreceptors are no porous sensilla.

L369-373, Sensilla campaniformia Sca.

L379, 371 : it is wrong : porous Sca are not sensilla campaniformia (see McIver 1975). All

sensilla campaniformia in the world are aporous and proprioceptors. The center of Sca in

Notobitus shows an ecdysial pore (moulting pore) and not a sensory pore (see Faucheux 1999).

They always possess an unique sensory neuron. These sensilla in Notobitus are

mechanoreceptors proprioceptors located to the joints.

L373-374, Bb are not described in Notobitus, they are microtrichia.

Response 3: I have modified them.

Point 4: References :

Write in italics all scientific names and scientific journals.

Figures : good quality

Response 4: I have examined the entire references and corrected them.

Reviewer 3 Report

The authors should make an effort to be more concise. The terminology used should be uniform along all text. The terms used to describe the morphological characteristics of antennae receptors (e.g. Tables S2 and S3) should be reviewed. In some cases, their meaning (e.g. "Mortar") does not allow a good relation with the structure actually seems to be. 

Author Response

Response to Reviewer 3 Comments

Dear anonymous reviewer,

I have modified the manuscript according to your comments. Thank you very much for your valuable advice! The following is the responses to the questions:

Point 1: The authors should make an effort to be more concise. The terminology used should be uniform along all text. The terms used to describe the morphological characteristics of antennae receptors (e.g. Tables S2 and S3) should be reviewed. In some cases, their meaning (e.g. "Mortar") does not allow a good relation with the structure actually seems to be.

Response 1: I have modified my full text including legend to achieve concise and terminology uniform along all text. I looked up the literature and thought my expression was really bad, so I changed “Mortar” into “Tight” by referring to the following literature.

Yi, Z.; Liu, D.; Cui, X.et al. Morphology and ultrastructure of antennal sensilla in male and female Agrilus mali (Coleoptera: Buprestidae). J. Insect Sci. 2016, 16, 87.

Point 2: How many individuals of each nymphal instar were collected?

"the" adult; "an" adult? How many were collected? Please let it clear.

Response 2: I have added “five individuals of the adult and each nymphal instar of N. meleagris”. (L107)

Point 3: This terminology of sensilla was created by yourselves or by other authors. In any case, let it clear and in the latter case, please provide the respective sources.

Response 3: As written in Materials and Methods (L137), my identification and naming of sensors refer to previous studies.

Schneider D. Insect Antennae. Annu.ev. Entomol. 1964, 9, 103–122.

Zacharuk, R.Y. Innervation of Sheath Cells of an Insect Sensillum by a Bipolar Type II Neuron. Can. J Zool. 1980, 58, 1264–1276.

Point 3: The expression “protective other sensilla” is not clear.

Response 3: According to your suggestion, I also think this place is not very clear, so I delete it.

Look forward to hearing from you soon.

Thank you very much.

With best regards

Zhu Wenli

Round 2

Reviewer 1 Report

From what I see, my suggestions were not taken into consideration almost at all.

You did not change the title according to suggestions. You do not show ultrastructure, therefore the word "ultrastructure" has to be deleted.

As for the references about antennal sensilla - you added TWO references and did not even cite it in the text. This is not "extensive". Antennal sensilla were studied in many families of Nepomorpha, in Gerromoprha, Reduviidae, Miridae, Alydidae, Pentatomidae, Coreidae and others. And EXTENSIVE list of references is required to show that you have actually done research about the subject you present.

You still did not provide any morphological description of the sensilla you describe. Why are they called trichodea, chaetica, basiconica, etc? What morphological characteristics imply that this is that type? Do they have pores? What type of socket they have? Are their surface ribbed or smooth? 

The manuscript still requires major revision.

Author Response

Response to Reviewer 1 Comments

Dear anonymous reviewer,

I have modified the manuscript according to your comments. Thank you very much for your valuable advice! The following is the responses to the questions:

Point 1: From what I see, my suggestions were not taken into consideration almost at all.

Response 1: Thank you again for the valuable advice of my manuscripts. Each of your suggestions has been modified after the discussion of all the authors, but some problems may not be well solved, so we have made modifications to these problems. Could you please check whether these modifications are OK?

Point 2: You did not change the title according to suggestions. You do not show ultrastructure, therefore the word "ultrastructure" has to be deleted.

Response 2: I misunderstood your suggestion before, now I have deleted ultrastructure.

Point 3: As for the references about antennal sensilla - you added TWO references and did not even cite it in the text. This is not "extensive". Antennal sensilla were studied in many families of Nepomorpha, in Gerromoprha, Reduviidae, Miridae, Alydidae, Pentatomidae, Coreidae and others. And EXTENSIVE list of references is required to show that you have actually done research about the subject you present.

Response 3: I am very sorry that I did forget to quote these in the article, and I have added these now. Maybe my expression is wrong. What I want to express in this sentence is that other scholars have done a lot of relevant research, and I have adjusted this sentence now.

Point 4: You still did not provide any morphological description of the sensilla you describe. Why are they called trichodea, chaetica, basiconica, etc? What morphological characteristics imply that this is that type? Do they have pores? What type of socket they have? Are their surface ribbed or smooth?

Response 4: I have written all of the description of sensilla in the results and Table S2、Table S3. Such as L182, 194.

As written in Materials and Methods (L137), my identification and naming of sensors refer to previous studies.

Schneider D. Insect Antennae. Annu.ev. Entomol. 1964, 9, 103–122.

Zacharuk, R.Y. Innervation of Sheath Cells of an Insect Sensillum by a Bipolar Type II Neuron. Can. J Zool. 1980, 58, 1264–1276.

Look forward to hearing from you soon.

Thank you very much.

With best regards

Zhu Wenli

Reviewer 2 Report

Microtrichia are not sensilla because they are noinnervated

Microtrichia sensilla don't exist.

Delete L355-357 and references 29, 47, 48 (only for ringe of microtrichia of sensilla coeloconica)

Author Response

Response to Reviewer 2 Comments

Dear anonymous reviewer,

I have modified the manuscript according to your comments. Thank you very much for your valuable advice! The following is the responses to the questions:

Point 1: Microtrichia are not sensilla because they are noinnervated

Microtrichia sensilla don't exist.

Response 1: This is my wrong understanding. I have deleted the expression related to

Microtrichia sensilla.

Point 2: Delete L355-357 and references 29, 47, 48 (only for ringe of microtrichia of sensilla coeloconica)

Response 2: I have deleted the description of microtrichia in the discussion section and made adjustments to the three literatures

Look forward to hearing from you soon.

Thank you very much.

With best regards

Zhu Wenli